# Downregulation of LncRNA GCLC-1 Promotes Microcystin-LR-Induced Malignant Transformation of Human Liver Cells by Regulating GCLC Expression

**DOI:** 10.3390/toxics11020162

**Published:** 2023-02-09

**Authors:** Xinglei Huang, Zhaohui Su, Jiangheng Li, Junquan He, Na Zhao, Liyun Nie, Bin Guan, Qiuyue Huang, Huiliu Zhao, Guo-Dong Lu, Qingqing Nong

**Affiliations:** 1Department of Environmental Health, School of Public Health, Guangxi Medical University, Nanning 530021, China; 2Department of Clinical Laboratory, The Affiliated Tumor Hospital of Guangxi Medical University, Nanning 530021, China; 3Department of Toxicology, School of Public Health, Guangxi Medical University, Nanning 530021, China; 4Guangxi Colleges and Universities Key Laboratory of Prevention and Control of Highly Prevalent Diseases, Guangxi Medical University, Nanning 530021, China; 5Guangxi Key Laboratory of Environment and Health Research, Guangxi Medical University, Nanning 530021, China

**Keywords:** microcystin-LR, lncRNA GCLC-1, GCLC, carcinogenesis

## Abstract

Microcystin-LR (MCLR) is an aquatic toxin, which could lead to the development of hepatocellular carcinoma (HCC). Long non-coding RNAs (lncRNAs) are considered important regulatory elements in the occurrence and development of cancer. However, the roles and mechanisms of lncRNAs during the process of HCC, induced by MCLR, remain elusive. Here, we identified a novel lncRNA, namely lnc-GCLC-1 (lncGCLC), which is in close proximity to the chromosome location of glutamate–cysteine ligase catalytic subunit (GCLC). We then investigated the role of lncGCLC in MCLR-induced malignant transformation of WRL68, a human hepatic cell line. During MCLR-induced cell transformation, the expression of lncGCLC and GCLC decreased continuously, accompanied with a consistently high expression of miR-122-5p. Knockdown of lncGCLC promoted cell proliferation, migration and invasion, but reduced cell apoptosis. A xenograft nude mouse model demonstrated that knockdown of lncGCLC promoted tumor growth. Furthermore, knockdown of lncGCLC significantly upregulated miR-122-5p expression, suppressed GCLC expression and GSH levels, and enhanced oxidative DNA damages. More importantly, the expression of lncGCLC in human HCC tissues was significantly downregulated in the high-microcystin exposure group, and positively associated with GCLC level in HCC tissues. Together, these findings suggest that lncGCLC plays an anti-oncogenic role in MCLR-induced malignant transformation by regulating GCLC expression.

## 1. Introduction

Microcystins (MCs), a group of cyclic heptapeptide compounds, are secondary metabolites produced by freshwater cyanobacteria [1]. So far, over 279 structural variants of MCs have been identified, among which microcystin-LR (MCLR) is regarded as the most toxic [2]. Since current tap water treatment technologies are not effective in removing MCs, MCs contamination has become an imperative threat to people’s drinking water safety. Epidemiological studies have suggested that chronic exposure to MCLR via consumption of contaminated water was associated with an increased incidence of hepatocellular carcinoma (HCC) [3,4,5,6,7]. MCLR is currently classified as a group 2B carcinogen by the International Agency for Research on Cancer [8]. However, the underlying mechanism of MCLR-induced HCC remains elusive. 

One of the predominant forms of free radical-induced oxidative lesions, 8-hydroxy-2’-deoxyguanosine (8-OHdG), has been widely used as a biomarker for oxidative stress and carcinogenesis [9]. Long-term and persistent exposure to MCLR increased the 8-OHdG levels of DNA in liver cells, damaged the integrity of mitochondrial DNA (mtDNA) and nuclear DNA, and altered the mtDNA content [10,11]. A large number of proteins has been identified to be involved in the DNA damage process and response. Glutamate–cysteine ligase (GCL) is one of the key enzymes of oxidative stress and functions as a rate-limiting enzyme of glutathione (GSH) synthesis. This enzyme is a heterodimer, consisting of a catalytic subunit (GCLC) and a modifier subunit (GCLM) [12]. Downregulation of GCLC was observed in multiple types of human cancer cell lines and tumors [13,14,15]. Previously, we have found that the expression of GCLC and activity of GCL decreased continuously, accompanied with consistently low levels of GSH, but high levels of oxidative DNA damages during MCLR-induced cell transformation, suggesting that downregulation of GCLC participates in MCLR-induced oxidative DNA damages and malignant transformation in human liver cells [16].

Long non-coding RNAs (lncRNAs) are a group of non-coding transcripts, longer than 200 nucleotides. Abnormalities in lncRNAs have been confirmed to exhibit tumor suppressor or carcinogenic effects, and play an important role in the development of tumors. For instance, the expression of lncRNA-DQ786227 was significantly increased during the transformation of BEAS-2B cells induced by benzo(a)pyrene, while silencing of lncRNA-DQ786227 expression inhibited cell proliferation and colony formation, and promoted cell apoptosis [17]. LncRNA linc00152 was upregulated in transformed 16HBE cells, induced by cigarette smoke extract (CSE), and expedited cell transformation by regulating cyclin D1 [18]. Cadmium (Cd) exposure-caused lncRNA MEG3 downregulation leads to enhanced cell cycle progression and apoptosis resistance, promoting Cd-induced cell transformation and cancer stem cell (CSC)-like property [19]. Furthermore, numerous lncRNAs, including lncRNA-LET [20], lncRNA-PRAL [21] and lnc-DILC [22] were found to be downregulated in HCC tissues and dramatically inhibited HCC growth by regulating histone acetylation, p53 ubiquitination, or autocrine interleukin-6/STAT3 signaling. However, the roles of lncRNAs, in the occurrence of cancer induced by MCLR, are poorly reported.

It is well established that the function of lncRNAs is frequently associated with their chromosomal location. Accumulating evidence suggests that numerous lncRNA loci act locally (in cis) to regulate the expression of neighbouring genes through functions of the lncRNA promoter, transcription, or transcript itself [23]. Recently, we identified a novel lncRNA named lnc-GCLC-1 (defined as lncGCLC here: https://lncipedia.org/db/transcript/lnc-GCLC-1:7, accessed on 20 December 2022), of which the chromosome location (chr6: 53,561,289–53,617,007) is near that of GCLC (chr6: 53,497,341–53,616,970). To date, the biological function and expression of lncGCLC remains unexplored. In this study, we investigated the potential roles of lncGCLC in MCLR-induced hepatocarcinogenesis, based on the previously established MCLR-induced malignant transformation model in human liver cell lines WRL68 and the MC-exposed human samples. Further, we analyzed the relationship among lncGCLC, redox and hepatocarcinogenesis induced by MCLR.

## 2. Materials and Methods

### 2.1. Cell Lines and MCLR-Induced Malignant Transformation Model

The human normal liver cell lines WRL68 and human hepatoma cell lines (HepG2 and SMMC7721) were purchased from the American Type Culture Collection (Manassas, VA, USA). WRL68, HepG2 and SMMC7721 cells were cultured in Dulbecco’s Modified Eagle’s Medium (DMEM, Gibco, USA), supplemented with 10% fetal bovine serum (FBS, Gibco, USA) and 1% penicillin/streptomycin (Gibco, USA) in a cell incubator, with 5% carbon dioxide at 37 °C. The MCLR-induced malignant transformation model of WRL68 cells was established as described in the previous study [24,25]. In brief, the cells were continuously exposed to phosphate-buffered saline (PBS; negative control) or 10 μg/L of MCLR (Alexis, Switzerland) for 72 h per passage. The process was continued for about 11 weeks (25 passages). The soft agar assay and in vivo tumorigenicity assays in nude mice were performed to detect the malignancy of MCLR-treated WRL68 cells [16].

### 2.2. Construction of WRL68 Cells with Stable Knockdown of LncGCLC

Small hairpin RNA (shRNA) was synthesized against lncGCLC by iGeneBio (Guangzhou, China). The shRNA, or scrambled control shRNA of lncGCLC, was cloned into vector psi-LVRH1GPF, named sh-lncGCLC and sh-NC, respectively. Then, the constructed plasmids were transfected into 293T cells, to collect viral particles using Lenti-Pac™ HIV packaging mix (GeneCopoeia, Inc., Guangzhou, China). The viral particles were then used to infect WRL68 cells, and screened by adding puromycin (10 μg/mL). The stably transfected cell line was confirmed by a fluorescence microscope and quantitative real-time polymerase chain reaction (qRT-PCR). The interference sequence of sh-lncGCLC and sh-NC are as follows: sh-lncGCLC, 5′-GCTTCTCCTCACTCCCAATTA-3′; sh-NC, 5′-GCTTCGCGCCGTAGTCTTA-3′. Then, the vector, or sh-lncGCLC-transfected cells, was separately treated with PBS or 10 μg/L of MCLR for 25 passages.

### 2.3. Tumor Tissue of HCC Patients with MC Exposure

HCC patients with an MC exposure were selected from our previous case-control study that involved 541 participants [26]. Briefly, thirty pairs of liver tumor tissue and adjacent normal tissue were collected during surgery on patients diagnosed with HCC from the Affiliated Tumor Hospital of Guangxi Medical University. Tissue samples were used to detect the expressions of lncGCLC and GCLC using qRT-PCR. Serum MC levels were determined using enzyme-linked immunosorbent assay (ELISA) kits (Beacon Analytical Systems Inc., Saco, ME, USA). The 30 participants were matched by age, sex, and hepatitis B virus infection status, and were divided into two groups according to the median values of serum MCs (0.14 μg/L) from all 541 HCC patients, termed the low MC exposure group and the high MC exposure group, respectively. Informed consent was obtained from patients prior to specimen collection, and all samples were snap-frozen in liquid nitrogen, immediately after excision, and stored at −80 °C until use. 

### 2.4. RNA Extraction and qRT-PCR Analysis

Total RNA was isolated from cells and tissues using a TRIzol reagent (Invitrogen, Carlsbad, CA, USA), according to the manufacturer’s instructions. RNA quality and concentration were measured using NanoDrop ND-1000 Spectrophotometer (Agilent, Santa Clara, CA, USA). Total RNA was reverse-transcribed using a PrimeScript1 RT Reagent Kit (TaKaRa, Kusatsu, Japan), and the SYBR Premix Ex Taq^TM^ Kit (TaKaRa, Kusatsu, Japan) was then used to determine gene expression with gene-specific primers, according to the manufacturer’s instructions. PCR reactions were performed on the StepOnePlus^TM^ Real-Time PCR System (Applied Biosystems, Foster City, CA, USA), and the cycling conditions were as follows: 95 °C for 30 s, followed by 40 cycles of 95 °C for 5 s and 60 °C for 30 s. The PCR products were identified using a melting-curve analysis. Relative fold changes were calculated using the 2^−△△Ct^ method. The sequences of primers are shown in Appendix A.

### 2.5. Cell Proliferation Assay

A cell proliferation assay was performed using a Cell Counting Kit-8 (CCK-8; Dojindo, Japan), abiding by the manufacturer’s protocols. Cells were seeded into 96-well plates at 5 × 10^3^ cells per well. After incubation for 24, 48, 72, or 96 h, the culture medium was discarded and 110 μL of fresh medium containing 10 μL of CCK-8 reagent was added to each well. After incubation for 2 h, A450 was measured using a microplate reader (Thermo Fisher Scientific, Rochester, NY, USA).

### 2.6. Flow Cytometry Assay

For cell cycle analysis, cells were harvested and fixed overnight in 70% (*v*/*v*) ethanol at 4 °C. Subsequently, cells were stained with 1 mg/mL propidium iodide at 37 °C for 30 min after mixing with 10 mg/mL RNase and assessed immediately by flow cytometry (FACScan; BD Biosciences, Shanghai, China). To analyse cell apoptosis, cells were harvested and washed twice with ice-cold PBS, followed by resuspension in the binding buffer. Subsequently, 5 μL of Annexin V-APC and 10 μL of 7-AAD were added to 100 μL of the cell suspension at 4 °C for 15 min in the dark. Cell apoptosis was determined by flow cytometry (FACScan; BD Biosciences, Shanghai, China). Data on the cell cycle and apoptosis were analysed by FlowJo 7.6 (Treestar, Woodburn, OR, USA).

### 2.7. Transwell Migration and Invasion Assays

Migration or invasion of cells in each group was evaluated using 24-well Transwell chambers (BD Biosciences, Becton, NJ, USA) without or with Matrigel-coating, according to the manufacturer’s instructions. Cells were harvested and adjusted to 1 × 10^5^/mL of cell suspension using serum-free medium, containing 0.08% bovine serum albumin. Briefly, 200 μL cell suspension was added to the upper chamber of the Transwell (8-μm pore size). Subsequently, 2 × DMEM medium was poured into the bottom well. After incubating at 37 °C for 60 h, cells were fixed with 4% paraformaldehyde, and stained with 0.1% crystal violet. After removing non-migrated or non-invaded cells from the top of the membrane with cotton swabs, migrated or invaded cells from the bottom of the membrane were counted using an inverted microscope (Olympus, Tokyo, Japan), in three randomly selected fields (magnification; 100×).

### 2.8. Soft Agar Assay

The cells in the logarithmic growth phase were harvested and adjusted to a density of 5 × 10^4^/mL. The agar (1.2% *w*/*v*, Sigma-Aldrich, St. Louis, MO, USA), in a small beaker, was completely dissolved using a microwave oven, cooled to 60 °C, and rapidly mixed with the same volume of 2 × DMEM medium (containing 20% FBS). The mixture was then injected into each well of 6-well plates. After solidification at room temperature, 5 × 10^3^ cells per well of each group were suspended in 2 mL of 0.3% (*w*/*v*) agar medium, supplemented with 10% FBS and then seeded in the plate (three plates per group). These plates were incubated at 37 °C under 5% CO_2_ for 15 days and stained with 0.005% crystal violet for 30 min. The colony formation efficiency was measured when clones contained more than 50 cells under an inverted microscope (Olympus, Tokyo, Japan). The rate of soft agar colony formation was calculated as follows: (number of colonies)/(number of cells seeded) × 100%.

### 2.9. Tumorigenicity Assays in Nude Mice

Thirty-six BALB/c athymic nude mice (4–5 weeks, male: female = 1:1) were purchased from the Guangxi Medical University Laboratory Animals Center. After a week of adaptive feeding in a pathogen-free room, mice were randomly divided into six groups (n = 3 for both male and female nude mice per group). The 25th passage of sh-lncGCLC-transfected cells, vector-transfected cells, and control cells (with or without MCLR treatment) were harvested after adjusting the cell suspension to 1 × 10^7^/mL with PBS, then, 100 μL was injected subcutaneously into nude mice to evaluate the tumorigenicity of transformed cells. The tumor volume was measured every three days and calculated using the equation: V = (LW^2^)/2 (L, longitudinal diameter; W, width diameter) [25,27]. After 22 days of injection, the mice were sacrificed, and tumors were harvested, weighed, and fixed with 4% paraformaldehyde for pathological examination.

### 2.10. Relative Expression of LncGCLC in the Nucleus and Cytoplasm

To establish the subcellular localization of lncGCLC, nuclear and cytoplasmic fractions were isolated from WRL68 cells using the Nuclear/Cytosol Fractionation Kit (BioVision, Milpitas, MA, USA), and nuclear and cytoplasmic RNA was extracted. Nuclear and cytoplasmic RNA (800 ng) was then converted to cDNA and analyzed for the expression of lncGCLC by qRT-PCR.

### 2.11. Western Blots Analysis

Briefly, cells were harvested and lysed using the RIPA buffer (Beyotime, Shanghai, China). Total proteins were collected from the supernatant liquid of the cell homogenate after centrifuging, and the concentrations were measured by a BCA protein assay kit (Beyotime, Shanghai, China). The proteins were subjected to sodium dodecyl sulfate-polyacrylamide gel electrophoresis (SDS-PAGE) and were electrophoretically transferred to nitrocellulose filter membranes. Then, the membranes were blocked in 5% fat-free milk for 2 h and incubated with the appropriate primary antibodies overnight at 4 °C with gentle shaking. The primary antibodies used were GCLC (1:400, Abcam, Cambridge, UK) and β-actin (1:5000, Abcam, Cambridge, UK). Subsequently, the membranes were incubated with secondary antibodies (1:10^4^; Beijing Bioss Biotechnology Co., Ltd., Beijing, China) for 2 h at room temperature. Finally, the protein bands were determined by Odyssey two-color infrared fluorescence imaging system (Li-cor, Ltd., Lincoln, NE, USA). 

### 2.12. Determination of GCL, GSH, and 8-OHdG

The activity of GCL and the content of GSH were determined by the kits obtained from Nanjing Jiancheng Biology Engineering Institute (Nanjing, China). The content of 8-OHdG was measured by an ELISA kit, supplied by BlueGene Biotech Co., Ltd. (Shanghai, China). All procedures were performed according to the corresponding manufacturer’s protocols.

### 2.13. Statistical Analysis

All statistical analyses were performed using SPSS 19.0 software (IBM Corp., Armonk, NY, USA). Quantitative data are presented as the mean ± standard deviation (SD). Differences between any two groups were analyzed by a paired *t*-test or an independent *t*-test. Differences among multiple groups were assessed using one-way analysis of variance analysis (ANOVA). If the ANOVA was statistically significant, the student-Newman-Keuls test would be used to analyze the differences between the two groups. A value of *p* < 0.05 was statistically significant.

## 3. Results

### 3.1. LncGCLC Is Downregulated in MCLR-Transformed WRL68 Cells and HCC Tissues with MC Exposure

We investigated whether lncGCLC is differentially expressed in MCLR-transformed cells, liver cancer cell lines, and patient samples. Continuously decreased levels of lncGCLC were observed in WRL68 cells after exposure to MCLR for 10 passages, compared to the passage-matched control cells (Figure 1A). Using qRT-PCR, we examined the expression of lncGCLC in two cell lines derived from human liver cancers (HepG2 and SMMC7721). LncGCLC expression of the two liver cancer cell lines was much lower than that in normal human liver cell lines WRL68 (Figure 1B). Next, the detection of lncGCLC in liver tissue of HCC patients with MC exposure was performed to further determine the relationship between MC exposure and lncGCLC expression. The results are shown in Figure 1C,D; the lncGCLC expression of tumor tissue was significantly lower than that of the matched normal tissues. Compared to the low MC exposure group, the lncGCLC expression of tumor tissue was significantly downregulated in high MC exposure group. As shown in Appendix A, there was no significant difference between the groups divided by the clinicopathologic factors, including tumor size, differentiation, lymph node metastasis, and Barcelona Clinic Liver Cancer (BCLC) stages. 

### 3.2. Knockdown of lncGCLC Promoted MCLR-Induced Cell Proliferation

To further clarify the role of lncGCLC in MCLR-induced malignant transformation, lncGCLC was continuously knocked down in WRL68 cells (passage 0) using shRNA. Because lncGCLC is a 3653-bp non-coding RNA, the success rate of construction of lentiviral vector overexpressing lncGCLC is relatively low, thus knockdown of lncGCLC was used to confirm its function. As shown in Figure 2A, a low expression level of lncGCLC was detected in the WRL68 cells stably transfected with lncGCLC shRNA (sh-lncGCLC), using qRT-PCR. In contrast, there was no significant difference in the expression level of lncGCLC between the control cells and empty vector-transfected cells. Next, the vector- or sh-lncGCLC-transfected cells were separately exposed to 10 μg/L of MCLR for 25 passages. MCLR downregulated the expression of lncGCLC in the empty vector-transfected cells, and the alteration was more pronounced by lncGCLC knockdown. CCK-8 assays indicated that the proliferation rate of the MCLR-treated cells and lncGCLC knockdown cells at 25 passages were significantly faster than that of the control (Figure 2B). Moreover, knockdown of lncGCLC significantly promoted cell proliferation induced by MCLR. These results further proved the effect of lncGCLC knockdown on cell proliferation.

Because lncGCLC affects cell proliferation, we hypothesized that it may do so by affecting the cell cycle and/or apoptosis. The apoptotic rate in sh-lncGCLC-transfected cells, following treatment with MCLR, was significantly lower than in vector-transfected cells treated with MCLR, whereas the frequency of apoptosis in vector-transfected cells was similar to that of the control cells (Figure 2C,D). Knockdown of lncGCLC significantly inhibited cell apoptosis. As shown in Figure 2E,F, the percentage of WRL68 cells in the G0/G1 phase was significantly reduced, while that in the G2/M phases was significantly increased after treatment with MCLR for 25 passages. Examination of cell cycle changes in sh-lncGCLC-transfected cells, following treatment with MCLR, disclosed an evident increase in the percentage of cells in the S phase and a concomitant decrease in the percentage of cells in the G0/G1 and G2/M phases. These results suggested that lncGCLC affected cell proliferation by functioning as a cell cycle checkpoint.

### 3.3. Knockdown of lncGCLC Promoted MCLR-Induced Cell Migration and Invasion

To determine whether lncGCLC has any effect on the metastatic capacity of MCLR-induced malignant transformed cells, we examined the effects of lncGCLC knockdown on the motility of MCLR-untreated or -treated cells with a Transwell assay. The Transwell experiment found that the number of cells passing through the transwell membranes significantly increased after knockdown of lncGCLC (Figure 3A–D). These results further indicated that the expression of lncGCLC could promote the migration and invasion of malignantly transformed WRL68 cells induced by MCLR.

### 3.4. Knockdown of lncGCLC Promoted the Growth of Malignantly Transformed WRL68 Cells, Induced by MCLR in Nude Mice

The results of the colony forming assay experiment showed that anchorage -independent growth capacity of MCLR-transformed cells was significantly increased after knockdown of lncGCLC (Figure 4A,B). We performed a tumor xenograft experiment in nude mice to verify whether lncGCLC can promote the growth of tumor in vivo. In the group injected with sh-lncGCLC-transfected cells, MCLR-treated cells, vector-transfected cells with MCLR treatment or sh-lncGCLC-transfected cells with MCLR treatment, tumor incidence was 100%. In contrast, in the group injected with control cells or vector-transfected cells, none of the mice developed tumors, with 0% of tumor incidence. Moreover, the tumors of the mice injected with sh-lncGCLC-transfected cells, following treatment with MCLR, were significantly larger than those with vector-transfected cells, following treatment with MCLR (Figure 4C–F). This promoted the effect of lncGCLC knockdown on tumor growth consistent with our in vitro results. Tumors in nude mice induced by MCLR-transformed cells were composed of poorly differentiated hepatocytes (Figure 4G). Though the tumors tended to grow faster in male compared to female mice in the group injected with MCLR-transformed cells, the differences did not reach statistically significant level (Appendix A).

### 3.5. lncGCLC Regulates the Expression of GCLC in MCLR-Induced Malignant Transformation

Following the observation of the role of lncGCLC in MCLR-induced malignant transformation of WRL68 cells, we attempted to analyze the underlying mechanism. Recent studies have found that some lncRNAs act in cis, therefore they can regulate the expression of one or more nearby genes on the same chromosome [23,28]. We investigated whether lncGCLC acts in cis. The expression of five nearby genes, GCLC, ELOVL5, TMEM14A, ICK, and FBXO9, extending across approximately 1.5 Mb downstream of lncGCLC, was detected by qRT-PCR after the knockdown of lncGCLC by shRNA. As shown in Figure 5A, only GCLC mRNA expression in sh-lncGCLC-transfected cells was significantly lower than that in vector control cells, suggesting that lncGCLC may regulate GCLC expression. 

We also determined the cytoplasmic/nuclear distributions of lncGCLC. As shown in Figure 5B, the lncGCLC was located both in the cytoplasm and the nucleus, and the expression level in the cytoplasm was higher than that in the nucleus, indicating that lncGCLC may function as a regulator at the post-transcriptional level. Bioinformatics analysis employed to predict miRNAs that potentially interact with lncGCLC (http://starbase.sysu.edu.cn and http://bioinfo.life.hust.edu.cn/LNCediting, accessed on 20 December 2022) revealed direct interactions with miR-122-5p. 

To further validate interactions among lncGCLC, miR-122-5p and GCLC, we analyzed lncGCLC and GCLC expression patterns in HCC tissue with MC exposure. Our data suggested that lncGCLC and GCLC were lowly expressed in hepatocellular carcinoma but also positively correlated (Figure 5C). The GCLC expression significantly decreased in tumor tissue compared with normal liver tissue from HCC patients with MC exposure (Figure 5D). Furthermore, the GCLC expression in the high MC exposure group was significantly lower than those in the low MC exposure group (Figure 5E). With prolonged exposure time, expression of miR-122-5p increased and that of GCLC mRNA decreased in MCLR-treated cells (Figure 5F). Western blot data showed that GCLC protein expression was gradually decreased with cell passage during MCLR-induced malignant transformation (Figure 5G,H). As shown in Figure 5I–L, the knockdown of lncGCLC significantly upregulated the expression of miR-122-5p and suppressed MCLR-induced inhibition of GCLC at both mRNA and protein levels.

### 3.6. Knockdown of lncGCLC Reduces GSH Levels and Enhances Oxidative DNA Damages

To further test whether knockdown of lncGCLC is involved in redox regulation, we measured the GCL activity, GSH, and 8-OHdG contents in vector- or sh-lncGCLC-transfected cells exposed to MCLR for 25 passages. As shown in Figure 6A,B, continuous treatment of MCLR for 25 passages led to an evident decrease in GCL activities and GSH levels in the empty vector-transfected cells. On the contrary, the 8-OHdG levels were increased after exposure to MCLR for 25 passages (Figure 6C). Knockdown of lncGCLC significantly reduced GCL activities and GSH levels, and increased 8-OHdG levels.

## 4. Discussion

Although many lncRNAs have been reported to link with liver cancer development, little is known about their roles in MCLR-induced cell transformation. In this study, we conducted bioinformatics analysis and identified lncGCLC, a 3653 bp transcript without protein-coding potency, located on chromosome 6p12.1 (chr6: 53561289-53617007), upstream of the GCLC gene. Zhou et al. [29] previously reported that nickel exposure downregulated the expression of lncRNA MEG3 in a time- and dose-dependent manner, and low lncRNA MEG3 expression was shown to promote the malignant transformation of human bronchial epithelial cells induced by nickel. Reportedly, lncRNA-Dreh, lnc-DILC, and lncRNA-LET are poorly expressed in HCC cell lines and tissues, and act as tumor suppressors in the development of liver cancer [20,22,30]. We have shown that the expression level of lncGCLC in the liver cancer cell lines HepG2 and SMMC7721 was significantly lower than that in WRL68 cells. LncGCLC was significantly downregulated in HCC tissues compared with the adjacent non-HCC tissues. These results indicated that the downregulation of lncGCLC is a frequent event in liver cancer and lncGCLC may have an anti-oncogene-like function. In this study, we found that the expression of lncGCLC was continuously reduced during the process of MCLR-induced malignant transformation. The knockdown of lncGCLC caused a cell cycle arrest at the S Phase, inhibited cell apoptosis, and promoted cell proliferation, induced by MCLR. This finding implies that lncGCLC is involved in tumorigenesis through the inhibition of cell apoptosis and accelerated cell cycle progress, which is consistent with the knockdown of lncRNA MEG3, inhibiting apoptotic activity [31]. In addition, we found that knockdown of lncGCLC could effectively increase the capacities of invasion and migration of MCLR-transformed cells. More importantly, knockdown of lncGCLC could increase the degree of malignancy in vitro and effectively promote tumor growth in vivo. These results suggest that lncGCLC might play a pivotal role in the regulation of MCLR-induced malignant transformation. 

LncRNAs could act as sponges to compete miRNAs with mRNAs, participating in various biological processes [32]. For instance, lncRNA BCAR4 may promote the proliferation, migration, and invasion of liver cancer cells by directly binding to miR-1261 and targeting the anaphase-promoting complex subunit 11 (ANAPC11) gene [33]. GCLC, the most important subunit of GCL, contains all substrate-binding sites, and catalytic activity of GCL and plays an important role in GSH biosynthesis [34]. Mougiakakos et al. [14] reported that the malignant phenotype of melanoma cells, including survival, invasiveness, and switch from E-cadherin to N-cadherin expression, was found at significantly higher levels in cells with a lower GCLC expression. Previously, we have found that GCLC expression was progressively reduced during MCLR-induced cell transformation. Overexpression of GCLC decreased the capacities of proliferation, migration and invasion of MCLR-transformed cells [16]. In the present study, we found that lncGCLC exerted an inhibitory action on carcinogenesis, similar to that exhibited by GCLC. The expressions of lncGCLC and GCLC in human HCC tissue were significantly downregulated in the high MC exposure group. LncGCLC expression level showed a positive relation with GCLC levels in HCC tissue. During MCLR-induced cell transformation, the expression of lncGCLC decreased continuously, accompanied with consistent low expression of GCLC, but high expression of miR-122-5p. Knockdown of lncGCLC significantly upregulated miR-122-5p expression, and suppressed GCLC expression. MiR-122-5p is one of the most abundant miRNAs in the liver, constituting 70% of all hepatic miRNAs [35]. It has been reported that the expression of miR-122-5p was significantly higher in the alpha-fetoprotein (AFP)-producing gastric cancer tissues and plasma samples [36]. MiR-122-5p promotes aggression and epithelial–mesenchymal transition in triple-negative breast cancer, by suppressing charged multivesicular body protein 3 (CHMP3) through mitogen-activated protein kinase (MAPK) signaling [37]. Therefore, we speculate that lncGCLC may regulate GCLC expression by acting as an endogenous competitive RNA for miR-122-5p. Further studies are required to clarify these mechanisms.

Recently, it has been reported that long non-coding RNAs are involved in the regulation of redox reactions. Zhang et al. [38] revealed that downregulation of lncRNA MAGI2-AS3 decreased the H_2_O_2_ content and delayed cell senescence, by stabilizing the HSPA8 protein level. LncRNA HCP5 was found to promote the stemness and chemoresistance of gastric cancer cells by driving fatty acid oxidation [39]. LncRNA LCPAT1 was reported to be involved in DNA damage induced by cigarette smoke extract through the RCC2 gene [40]. It has been shown that the GCLC primarily regulates de novo synthesis of glutathione and is central to the antioxidant capacity of the cell [41]. In the present study, the levels of GCL and GSH were significantly decreased in WRL68 cells after exposure to the low concentration of MCLR for 25 passages. On the contrary, levels of 8-OHdG were increased after exposure to MCLR for 25 passages. Knockdown of lncGCLC significantly reduced GCL activities and GSH levels, and increased 8-OHdG levels, suggesting that lncGCLC may downregulate GCLC expression, reduce GSH levels, and subsequently induce oxidative DNA damage, which may contribute to oncogenesis caused by low-level MCLR. 

Among over 279 identified variants of MCs, MCLR is the most abundant and toxic. Several studies have also found that it is rather difficult to accurately determine the variants of MCs in human serum using the method of ELISA [2,42]. Currently, the commercial microcystin ELISA kit does not distinguish between MCLR and other MC variants (microcystin-RR, -YR, etc), but detects their presence at varying degrees. In this study, we used an ELISA method to detect serum MC levels for a rough estimation of MCLR exposure, which may not be able to accurately assess the internal exposure level of MCLR. Thus, there is a pronounced need to further investigate a more sensitive serum MCLR detection method and more appropriate biomarkers in exposed populations.

## 5. Conclusions

In conclusion, we identified a novel lncRNA, lncGCLC, which was downregulated in MCLR-transformed cells and tumor tissue of HCC patients. MCLR exposure-caused lncGCLC downregulation enhanced miR-122-5p expression, suppressed GCLC expression and GSH levels, and promoted oxidative DNA damages. Furthermore, the downregulation of lncGCLC enhanced cell proliferation, apoptosis resistance, migration and invasion, all of which promote malignant transformation of liver cells (Figure 7). More importantly, the expression of lncGCLC in human HCC tissues was significantly downregulated in the high MC exposure group, and positively associated with GCLC level in HCC tissues. Taken together, the current results suggest that lncGCLC plays an anti-oncogenic role in MCLR-induced malignant transformation by regulating GCLC expression. Thus, increasing the expression of lncGCLC by pharmacological agents may be a potential strategy to prevent the liver carcinogenesis of MCLR.

## Figures and Tables

**Figure 1 toxics-11-00162-f001:**
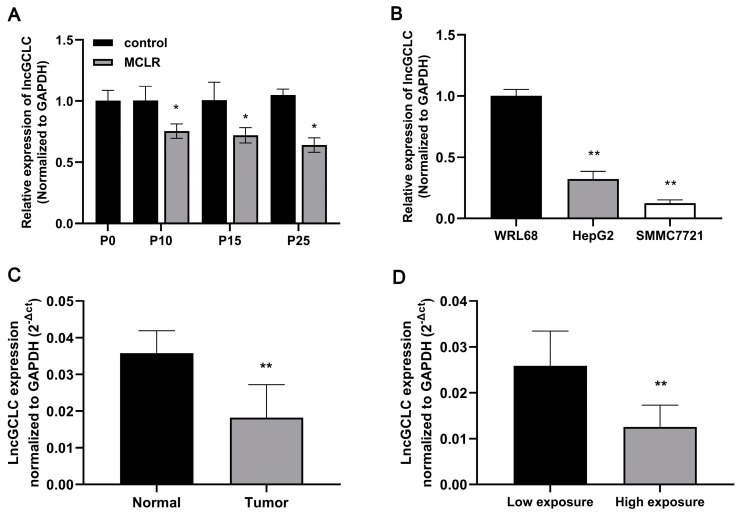
The lncGCLC expression in MCLR-exposed cells and population samples. (**A**) Changes of lncGCLC expression in WRL68 cells after exposure to 0 or 10 μg/L of MCLR for 0, 10, 15, and 25 passages. * *p* < 0.05 compared with passage-matched control cells. (**B**) Comparison of lncGCLC expression in WRL68, HepG2, and SMMC7721 cells. ** *p* < 0.01 compared with WRL68 cells. (**C**) Comparison of lncGCLC expression in tumor tissue and adjacent normal tissue from HCC patients with MC exposure (n = 30), ** *p* < 0.01. (**D**) LncGCLC expression in HCC patients with high MC exposure (serum MCs ≥ 0.14 μg/L, n = 17) was lower than that in those with low MC exposure (serum MCs < 0.14 μg/L, n = 13), ** *p* < 0.01. Data are presented as the mean ± SD of three independent experiments.

**Figure 2 toxics-11-00162-f002:**
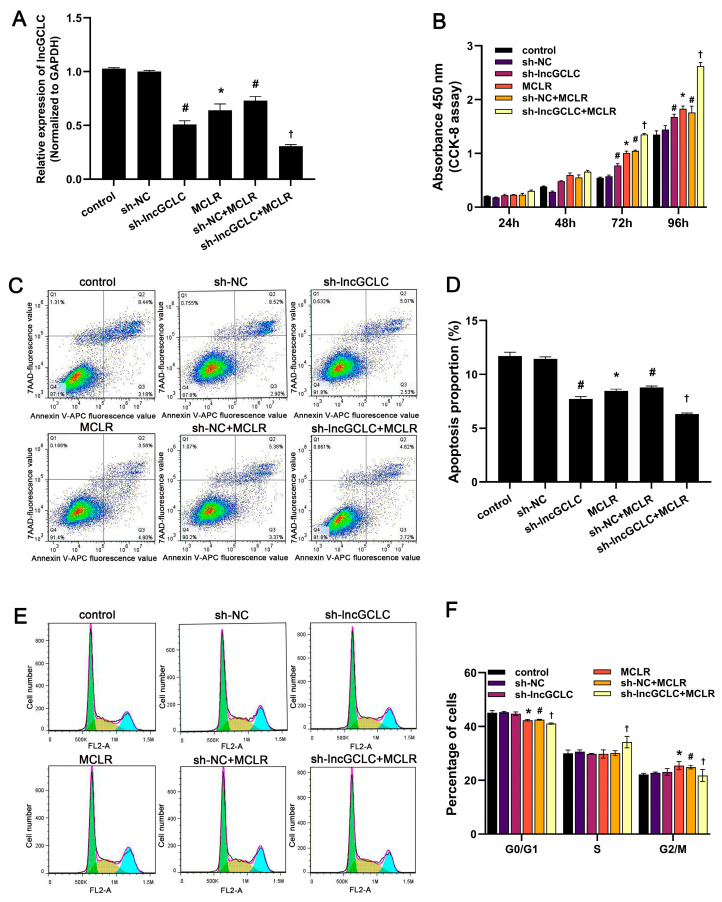
Knockdown of lncGCLC promoted the proliferation of MCLR-treated WRL68 cells. Vector- or sh-lncGCLC-transfected cells were treated with or without 10 μg/L of MCLR for 25 passages. (**A**) The expression of lncGCLC was detected using qRT-PCR. (**B**) Cell proliferation activity was detected by CCK-8. (**C**) Cell apoptosis was analyzed using flow cytometry. (**D**) Apoptosis rate in each group. (**E**) The percentage distribution of cells in the G1/G0, S, and G2/M phases of the cell cycle was determined by flow cytometry. (**F**) Cell cycle distribution quantification. Data are presented as the means ± SD of three independent experiments in each group. * *p* < 0.05 compared with the control group; ^#^
*p* < 0.05 compared with the sh-NC group; ^†^
*p* < 0.05 compared with the sh-NC + MCLR group.

**Figure 3 toxics-11-00162-f003:**
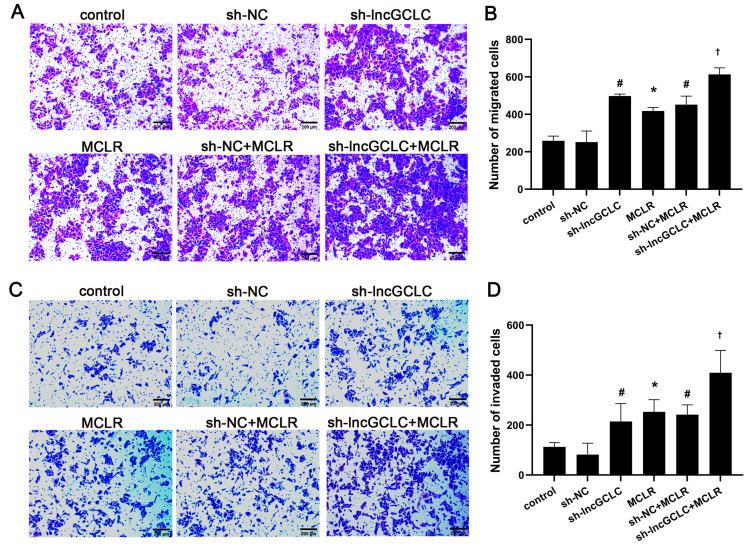
Knockdown of lncGCLC promoted the migration and invasion of MCLR-treated WRL68 cells. Vector- or sh-lncGCLC-transfected cells were treated with or without 10 μg/L of MCLR for 25 passages. (**A**) Representative images of a cell migration assay (100×), scale bar = 200 μm. (**B**) Quantification of cell migration. (**C**) Representative images of a cell invasion assay (Original magnification ×100, scale bar = 200 μm). (**D**) Quantification of cell invasion. Data are presented as the means ± SD of three independent experiments in each group. * *p* < 0.05 compared with the control group; ^#^
*p* < 0.05 compared with the sh-NC group; ^†^
*p* < 0.05 compared with the sh-NC + MCLR group.

**Figure 4 toxics-11-00162-f004:**
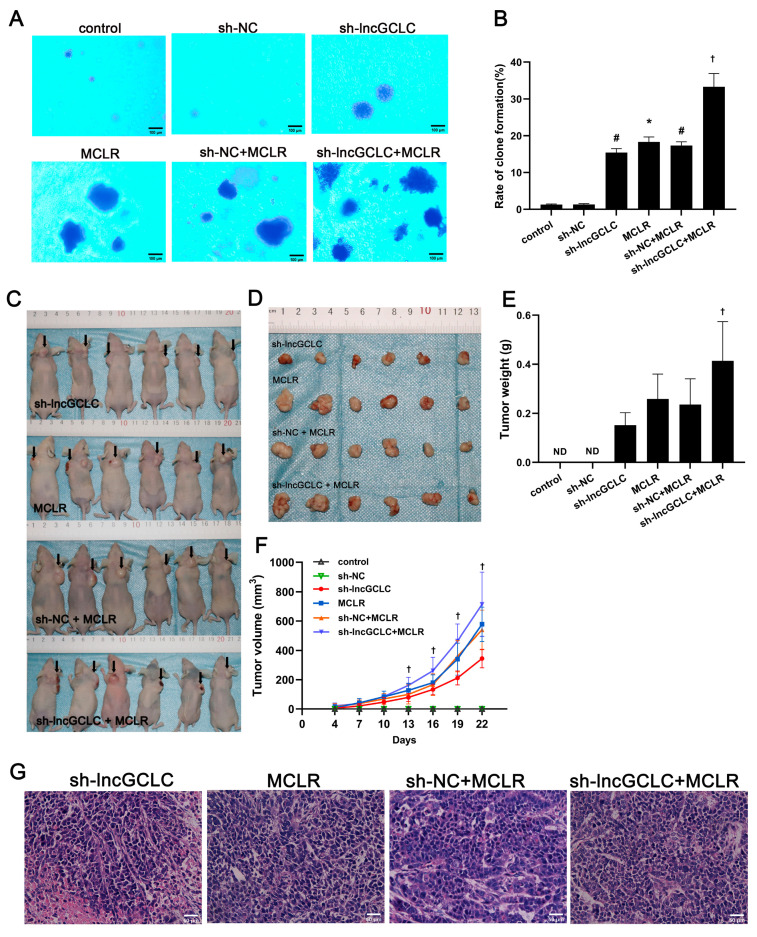
Knockdown of lncGCLC promoted the growth of MCLR-induced malignantly transformed WRL68 cells in nude mice. Vector- or sh-lncGCLC-transfected cells were treated with or without 10 μg/L of MCLR for 25 passages. (**A**) Representative images of a soft agar assay (Original magnification ×200, scale bar = 100 μm). (**B**) Quantification of colony formation in soft agar. (**C**) Tumorigenicity test of BALB/c nude mice was performed. Solid tumors were removed after the sacrifice at 22 days. (**D**) Representative image showing subcutaneous tumor size. (**E**) The weights of solid tumors. The values given are mean ± SD (n = 3 for both male and female nude mice per group). (**F**) Tumor volume was monitored every 3 days after injection of WRL68 cells. (**G**) Pathological changes of the tumor tissue in nude mice in each group (HE stains, original magnification ×400, scale bar = 50 μm). ND—not detected. Data are presented as the means ± SD of three independent experiments in each group. * *p* < 0.05 compared with the control group; ^#^
*p* < 0.05 compared with the sh-NC group; ^†^
*p* < 0.05 compared with the sh-NC + MCLR group.

**Figure 5 toxics-11-00162-f005:**
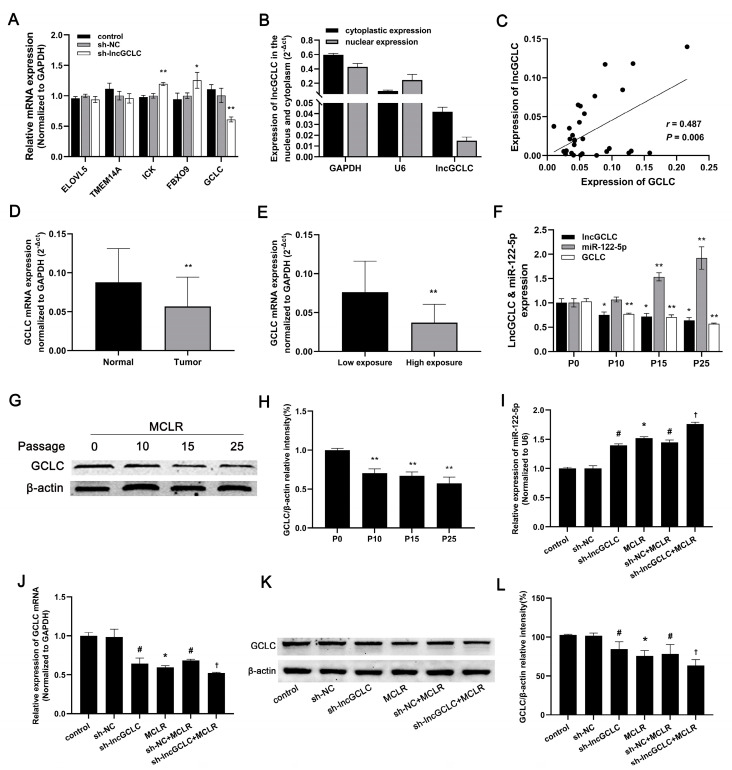
Verification of the interrelationships among lncGCLC, miR-122-5p, and GCLC. (**A**) The mRNA expression level of neighboring genes which are located nearly 1.5 Mb downstream of lncGCLC in lncGCLC knockdown cells. * *p* < 0.05 and ** *p* < 0.01 compared with vector control cells. (**B**) The expression level of lncGCLC in the nuclear and cytoplasmic fractions of WRL68 cells. (**C**) The correlation between the lncGCLC expression and the GCLC expression in HCC tissues (n = 30). Correlation coefficient (*r*) and *P* value were calculated by Pearson correlation analysis. (**D**) Comparison of GCLC expression in tumor tissue and adjacent normal tissue from HCC patients with MC exposure (n = 30), ** *p* < 0.01. (**E**) GCLC expression in HCC patients with high MC exposure (serum MCs ≥ 0.14 μg/L, n =17) was lower than in those with low MC exposure (serum MCs < 0.14 μg/L, n = 13), ** *p* < 0.01. (**F**) After exposure to 10 μg/L of MCLR for 0, 10, 15, and 25 passages, expression of lncGCLC, miR-122-5p and GCLC mRNA was detected in WRL68 cells. * *p* < 0.05 and ** *p* < 0.01 compared with passage-matched control cells. (**G**,**H**) Expression of GCLC protein in P0, P10, P15 and P25 MCLR-induced malignantly transformed WRL68 cells. ** *p* < 0.01 compared with passage-matched control cells. (**I**) Changes of miR-122-5p in both vector- and sh-lncGCLC-transfected WRL68 cells treated with 0 or 10 μg/L of MCLR for 25 passages. (**J**–**L**) Changes of GCLC mRNA (**J**) and protein (**K**,**L**) expression in both vector- and sh-lncGCLC-transfected WRL68 cells treated with or without 10 μg/L of MCLR for 25 passages. Data are presented as the means ± SD of three independent experiments in each group. * *p* < 0.05 compared with the control group; ^#^
*p* < 0.05 compared with the sh-NC group; ^†^
*p* < 0.05 compared with the sh-NC + MCLR group.

**Figure 6 toxics-11-00162-f006:**
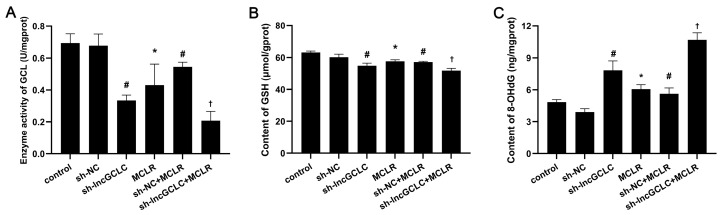
Knockdown of lncGCLC reduced GSH levels and induced oxidative DNA damage in MCLR-treated WRL68 cells. Alterations in GCL activity (**A**), GSH (**B**), and 8-OHdG content (**C**) in vector- or sh-lncGCLC-transfected cells exposed to 0 or 10 μg/L of MCLR for 25 passages. Data are presented as the means ± SD of three independent experiments in each group. * *p* < 0.05 compared with the control group; ^#^
*p* < 0.05 compared with the sh-NC group; ^†^
*p* < 0.05 compared with the sh-NC + MCLR group.

**Figure 7 toxics-11-00162-f007:**
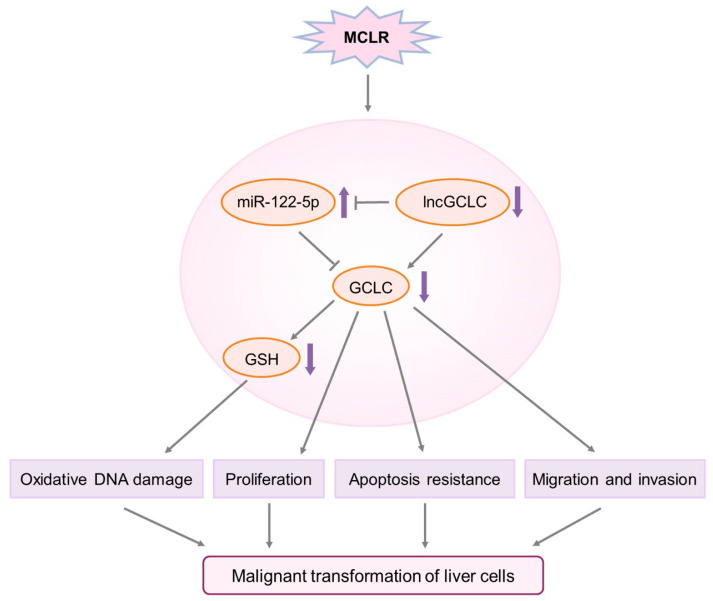
Schematic diagram for the role of lncGCLC pathway in MCLR-induced malignant transformation.

## Data Availability

The data are currently not publicly available, as the data also form part of an ongoing study.

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
