# Peer review of "Downregulation of LncRNA GCLC-1 Promotes Microcystin-LR-Induced Malignant Transformation of Human Liver Cells by Regulating GCLC Expression"

_toxics, 2023, doi:10.3390/toxics11020162_

Round 1

Reviewer 1 Report

Journal: Toxics (ISSN 2305-6304)

Manuscript ID: toxics-2143435

Type: Article

Title: Down-regulation of lncRNA GCLC-1 promotes microcystin-LR-induced malignant transformation of human liver cells through regulating GCLC expression

Topic: Environmental Toxicology and Human Health

This manuscript aimed to study the role of lncGCLC pathway in MCLR-induced malignant transformation. Assays of knockdown of lncGCLC, cell proliferation, invasion and migration, xenograft nude mice model, human HCC tissues, as wells as GCLC expression and GSH levels, oxidative DNA damages were performed. However, the writing and presentation are not good. I have the following comments and suggestions for the authors to improve the quality of the manuscript.

1. Abstract

Lines 24-27

“After establishing the transformed cell model using the human liver cell lines WRL68, we found that the lower expression of lncGCLC was continuously observed during MCLR-induced cell transformation, the lower expression of GCLC but the higher expression of miR-122-5p.”

The sentence is too long. Please divide it to two sentences.

2. Section “1. Introduction”

Lines 39-41

“So far, over 100 structural variants of MCs have been identified, among which microcystin-LR (MCLR) is regarded as the most toxic [2].”

More than 279 MC structural variants have been reported. Please read and cite the following paper.

Chen et al., 2021. Challenges of using blooms of Microcystis spp. in animal feeds: A comprehensive review of nutritional, toxicological and microbial health evaluation. https://doi.org/10.1016/j.scitotenv.2020.142319

3. Section “2. Materials and methods”

Line 104

“in vivo”

It should be italic.

4. Lines 123-125

“Serum MCLR levels were determined using enzyme-linked immunosorbent assay (ELISA) kits (Beacon Analytical Systems Inc, Saco, USA).”

More than 279 MC structural variants have been reported. Method by enzyme-linked immunosorbent assay (ELISA) cannot determine the variants of MCs. Please read and cite the following paper. Please change the related expressions “MC-LR” to “MCs” in the revised manuscript. Please add some discussion for the limitations.

Please read and cite the following paper.

Chen et al., 2021. Challenges of using blooms of Microcystis spp. in animal feeds: A comprehensive review of nutritional, toxicological and microbial health evaluation. https://doi.org/10.1016/j.scitotenv.2020.142319

5. Lines 227-229

“If ANOVA was statistically significant, the Student-Newman-Keuls test or the least significant difference would be used to analyze the differences between the two groups.”

The least significant difference analysis is not robust. Please use the Student-Newman-Keuls test.

6. Section “3. Results”

Lines 245-247

“There was no significant difference between the groups divided by the clinicopathologic factors, including tumor size, differentiation, histology type, lymph node metastasis, and TMN stage.”

Please present the results in figures in the supplementary file. Then readers can better understand your data.

7. Figures 2, 3, 4, 5 and 6

Markers of a , b and c for presentation of statistical differences among groups.

a P < 0.05 compared with control cells. b P < 0.05 compared with vector control cells. c P < 0.05 compared with vector control cells treated with MCLR

This is complex and confusing. Please use different letters to show significant differences among groups. For example, "a" and "b" or "bc" or "c" have significant differences, but "b" and "ab" or "bc" have no significant differences.

8. Figures 3AC

Please insert plotting scales to show the size of cells.

9. Figure 4A, G

Please insert plotting scales.

10. Figure 4, C-G

Please also present results of control and sh-NC.

11. Figure 5B

Please present results of quantitative, real expressions, but not percentages. Then readers can better understand your data.

12. Figure 5C

Similar to Figures 1 C and D, please also present results of GCLC transcriptions in tumor tissue and adjacent normal tissue from patients with MCLR exposure, as well as results in HCC patients with low and high MCLR exposure.

13. Please draw a figure to summarize role of lncGCLC pathway in MCLR-induced malignant transformation.

Author Response

Dear Editors and Reviewers:

Thank you for your letter and for the editors and reviewers’ comments concerning our manuscript entitled “Down-regulation of lncRNA GCLC-1 promotes microcystin-LR-induced malignant transformation of human liver cells through regulating GCLC expression”.

The comments are all valuable and helpful for us to revise and improve our manuscript. We have studied these comments carefully and have made modifications and corrections accordingly. Here we provide our responses, point by point, to the comments as listed below and highlighted the changes we have made in the revised manuscript.

We hope that the manuscript has been revised satisfactorily and will be acceptable for publication.

With best wishes,

Yours sincerely,

Qingqing Nong, MD, PhD

Professor of Preventive Medicine

School of Public Health

Guangxi Medical University

Nanning, 530021, P.R.China

Responds to the editors and reviewer’s comments:

Response to Reviewer 1

Comments and Suggestions for Authors:

This manuscript aimed to study the role of lncGCLC pathway in MCLR-induced malignant transformation. Assays of knockdown of lncGCLC, cell proliferation, invasion and migration, xenograft nude mice model, human HCC tissues, as wells as GCLC expression and GSH levels, oxidative DNA damages were performed. However, the writing and presentation are not good. I have the following comments and suggestions for the authors to improve the quality of the manuscript.

Response: Thank you for your suggestions. We feel very sorry that our writing and presentation are not good. We have revised the manuscript and asked some professional to help with scientific editing. The language errors throughout the whole paper have been corrected and marked in red in the revised manuscript.  

  1. Abstract

Lines 24-27

“After establishing the transformed cell model using the human liver cell lines WRL68, we found that the lower expression of lncGCLC was continuously observed during MCLR-induced cell transformation, the lower expression of GCLC but the higher expression of miR-122-5p.”

The sentence is too long. Please divide it to two sentences.

Response: Thank you for your suggestions. We feel very sorry that this sentence is really too long. According to your suggestion, we have divided it to two sentences and marked in red in the paper. (Abstract, line 23-26, page 1)

“We then investigated the role of lncGCLC in MCLR-induced malignant transformation in WRL68, a human hepatic cell line. During MCLR-induced cell transformation, the expression of lncGCLC and GCLC decreased continuously, accompanied with consistent high expression of miR-122-5p.”

  1. Section “1. Introduction”

Lines 39-41

“So far, over 100 structural variants of MCs have been identified, among which microcystin-LR (MCLR) is regarded as the most toxic [2].”

More than 279 MC structural variants have been reported. Please read and cite the following paper.

Chen et al., 2021. Challenges of using blooms of Microcystis spp. in animal feeds: A comprehensive review of nutritional, toxicological and microbial health evaluation. https://doi.org/10.1016/j.scitotenv.2020.142319

Response: Thank you for your suggestions. We are very sorry that we did not read the literature carefully. According to your suggestion, we have revised the sentence, cited this paper (Chen et al., 2021. Challenges of using blooms of Microcystis spp. in animal feeds: A comprehensive review of nutritional, toxicological and microbial health evaluation.), and marked in red in the revised manuscript. (Introduction section, line 39-41, page 1; Discussion section, line 472-480, page 13).

  1. Section “2. Materials and methods”

Line 104

“in vivo” It should be italic.

Response: Thank you for your suggestions. We have changed “in vivo” to italic type “in vivo” and marked in red in the paper. (Materials and methods section, line 3, page 102)

  1. Lines 123-125

“Serum MCLR levels were determined using enzyme-linked immunosorbent assay (ELISA) kits (Beacon Analytical Systems Inc, Saco, USA).”

More than 279 MC structural variants have been reported. Method by enzyme-linked immunosorbent assay (ELISA) cannot determine the variants of MCs. Please read and cite the following paper. Please change the related expressions “MC-LR” to “MCs” in the revised manuscript. Please add some discussion for the limitations.

Please read and cite the following paper.

Chen et al., 2021. Challenges of using blooms of Microcystis spp. in animal feeds: A comprehensive review of nutritional, toxicological and microbial health evaluation. https://doi.org/10.1016/j.scitotenv.2020.142319

Response: Thank you for your suggestions. We have changed the related expressions “MC-LR” to “MCs” in the revised manuscript, added some discussion for the limitations, and marked in red in the paper. (Materials and methods section, line 122-123, page 3 and line 126-127, page 3; Discussion section, line 472-480, page 13)

We also cited this paper (Chen et al., 2021. Challenges of using blooms of Microcystis spp. in animal feeds: A comprehensive review of nutritional, toxicological and microbial health evaluation.) and marked in red in the revised manuscript. (Discussion section, line 474, page 13).

  1. Lines 227-229

“If ANOVA was statistically significant, the Student-Newman-Keuls test or the least significant difference would be used to analyze the differences between the two groups.”

The least significant difference analysis is not robust. Please use the Student-Newman-Keuls test.

Response: Thank you for your suggestions. The least significant difference analysis is not robust. We are sorry to have ignored it. We used the Student-Newman-Keuls test to analyze the differences between the two groups again, revised the related description and marked in red in the revised manuscript. (Materials and methods section, line 224-125, page 5).

  1. Section “3. Results”

Lines 245-247

“There was no significant difference between the groups divided by the clinicopathologic factors, including tumor size, differentiation, histology type, lymph node metastasis, and TMN stage.”

Please present the results in figures in the supplementary file. Then readers can better understand your data.

Response: Thank you for your helpful suggestions. According to your suggestion, I added the Supplementary table 2. Clinicopathologic factors of hepatocellular carcinoma patients with MCs exposure.

We have changed “TMN stage” to “BCLC stage” and marked in red in the paper. (Results section, line 243-244, page 6)

  1. Figures 2, 3, 4, 5 and 6

Markers of a , b and c for presentation of statistical differences among groups.

a P < 0.05 compared with control cells. b P < 0.05 compared with vector control cells. c P < 0.05 compared with vector control cells treated with MCLR”

This is complex and confusing. Please use different letters to show significant differences among groups. For example, "a" and "b" or "bc" or "c" have significant differences, but "b" and "ab" or "bc" have no significant differences.

Response: Thank you for your helpful comments. Now, we changed "a", "b" or "c" to "*", "#" or "†" in order to show significant differences among groups.

For example, #P<0.05 compared with the sh-NC group, refers to the group marked with “#” was significantly different from the sh-NC group. (Results section, line 291-293, page 8; line 309-311, page 8; line 335-337, page 9-10; line 394-396, page 11; line 401-403, page 12).

  1. Figures 3AC

Please insert plotting scales to show the size of cells.

Response: Thank you for your suggestions. Now, we have insert plotting scales into Figures 3A and 3C.

  1. Figure 4A, G

Please insert plotting scales.

Response: Thank you for your suggestions. Now, we have insert plotting scales into Figure 4A and 4G.

  1. Figure 4, C-G

Please also present results of control and sh-NC.

Response: Thank you for your suggestions. We have added the experiment results of the control group and the sh-NC group in tumorigenicity assays. Figure 4C, 4D and 4G show representative pictures of subcutaneous tumor and pathological examination of the sh-lncGCLC group, MCLR group, sh-NC + MCLR group, sh-lncGCLC + MCLR group, respectively. There is no subcutaneous tumor in the control group and sh-NC group, thus experimental results of the control group and sh-NC group were shown in Figure 4E and 4F.

  1. Figure 5B

Please present results of quantitative, real expressions, but not percentages. Then readers can better understand your data.

Response: Thank you for your suggestions. As shown in Figure 5B, we have changed relative quantification of lncGCLC expression levels in the nuclear and cytoplasmic fractions to absolute quantification, using real expressions, but not percentages.

  1. Figure 5C

Similar to Figures 1 C and D, please also present results of GCLC transcriptions in tumor tissue and adjacent normal tissue from patients with MCLR exposure, as well as results in HCC patients with low and high MCLR exposure.

Response: Thank you for your suggestions. As shown in Figure 5D and 5E, we added the experimental results of GCLC transcriptions in tumor tissue and adjacent normal tissue from patients with MCLR exposure, as well as results in HCC patients with low and high MCLR exposure. We also explained the experimental results and marked in red in the paper (Results section, line 359-362, page 10).

  1. Please draw a figure to summarize role of lncGCLC pathway in MCLR-induced malignant transformation.

Response: Thank you for your suggestions. As shown in Figure 7, we draw a figure to summarize role of lncGCLC pathway in MCLR-induced malignant transformation.

Reviewer 2 Report

This paper aimed to investigate the role of lncGCLC in MCLR-induced cancer progression. The explanation of methodology and data presentation is done very well.

-Line 185: Please mention the gender of the mice.
-Figure legends are repeated. Please remove.
-Please check the manuscript for punctuation.
-Line 331: ...differentiated hepatocytes instead of hepatocellular?
-A schematic for the experimental design would be helpful.

-What is the therapeutic potential of lncGCLC in reducing MCLR-induced toxicity? Are there any compounds that work in correlation with them? Please discuss this in brief.

Author Response

Dear Editors and Reviewers:

Thank you for your letter and for the editors and reviewers’ comments concerning our manuscript entitled “Down-regulation of lncRNA GCLC-1 promotes microcystin-LR-induced malignant transformation of human liver cells through regulating GCLC expression”.

The comments are all valuable and helpful for us to revise and improve our manuscript. We have studied these comments carefully and have made modifications and corrections accordingly. Here we provide our responses, point by point, to the comments as listed below and highlighted the changes we have made in the revised manuscript.

We hope that the manuscript has been revised satisfactorily and will be acceptable for publication.

With best wishes,

Yours sincerely,

Qingqing Nong, MD, PhD

Professor of Preventive Medicine

School of Public Health

Guangxi Medical University

Nanning, 530021, P.R.China

Responds to the editors and reviewer’s comments:

Response to Reviewer 2

Comment: This paper aimed to investigate the role of lncGCLC in MCLR-induced cancer progression. The explanation of methodology and data presentation is done very well.

Response: We gratefully thank you for your positive comments and encouraging. The comments are all valuable and helpful for us to revise and improve our manuscript.

Suggestions from reviewer

-Line 185: Please mention the gender of the mice.

Response: Thank you for your suggestions. We have added a description of the gender of the mice and marked in red in the paper. (Materials and methods section, line 184-186, page 4)

“Thirty-six BALB/c athymic nude mice (4-5 weeks, male: female = 1:1) were purchased from the Guangxi Medical University Laboratory Animals Center.”

-Figure legends are repeated. Please remove.

Response: Thank you for your suggestions. We are very sorry for being so careless. Thank you for the reminder. We had removed duplicate Figure 1 legends.

-Please check the manuscript for punctuation.

Response: Thank you for your suggestions. We feel sorry for our carelessness. Now, we have revised the incorrect punctuation in the revised manuscript.

-Line 331: ...differentiated hepatocytes instead of hepatocellular?

Response: Thank you for your suggestions. We have changed “differentiated hepatocellular” to “differentiated hepatocytes” and marked in red in the paper. (Results section, line 324, page 9)

-A schematic for the experimental design would be helpful.

Response: Thank you for your suggestions.  As shown in Figure 7, we draw a schematic diagram for the experimental design.

-What is the therapeutic potential of lncGCLC in reducing MCLR-induced toxicity? Are there any compounds that work in correlation with them? Please discuss this in brief.

Response: Thank you for your suggestions. LncGCLC may be a novel and crucial biomarker in patients with HCC and can be used as a promising molecular target in reducing MCLR-induced toxicity. The current results suggest that increasing the expression of lncGCLC by pharmacological agents may be a potential strategy to prevent the liver carcinogenesis of MCLR. Because little is known about the relationship between lncGCLC and pharmacological agents, we discussed this in brief and marked in red in the paper. (Conclusions section, line 490-493, page 13).

Round 2

Reviewer 1 Report

Journal: Toxics (ISSN 2305-6304)

Manuscript ID: toxics-2143435-peer-review-v2

Type: Article

Title: Down-regulation of lncRNA GCLC-1 promotes microcystin-LR-induced malignant transformation of human liver cells through regulating GCLC expression

Topic: Environmental Toxicology and Human Health

This manuscript aimed to study the role of lncGCLC pathway in MCLR-induced malignant transformation. Assays of knockdown of lncGCLC, cell proliferation, invasion and migration, xenograft nude mice model, human HCC tissues, as wells as GCLC expression and GSH levels, oxidative DNA damages were performed. The revised manuscript improved a lot. I still have the following comments and suggestions for the authors to improve the quality of the manuscript.

1. Section “2. Materials and methods”

Lines 121-122

“Tissue samples were used to detect the expression of lncGCLC using qRT-PCR.”

Please change “the expression of lncGCLC” to “the expressions of lncGCLC and GCLC”.

2. Lines 124-127

“The 30 participants were matched by age, sex, and hepatitis B virus-infection status, and were divided into two groups according to the median values of serum MCs (0.14 μg/L), termed the low MCs exposure group and the high MCs exposure group, respectively.”

The 30 participants were divided into two groups according to the median values of serum MCs (0.14 μg/L). However, in the supplementary table S2, numbers of participants in termed the low MCs exposure group and the high MCs exposure group were 13 and 17, respectively. If according to the median values, then numbers in the two groups should be both 15. Please check the data.

3. Lines 183-185

“2.9. Tumorigenicity assays in nude mice

Thirty-six BALB/c athymic nude mice (4-5 weeks, male: female = 1:1) were purchased from the Guangxi Medical University Laboratory Animals Center.”

Figure 4, C-G

Both males and females were studied in this research. Were there any sex-dependent differences on tumor weight or volume? Please present both results of males and females.

4. Lines 185-187

“After a week of adaptive feeding in a pathogen-free room, mice were randomly divided into 6 groups (n=6 per group).”

It should be n=3 per group, for both males and females. Right? Please make the revisions in the revised manuscript.

5. Figures 1CD, 5CDE

Please present number of participants in each group in the revised manuscript.

6. Lines 242-244

“There was no significant difference between the groups divided by the clinicopathologic factors, including tumor size, differentiation, lymph node metastasis, and BCLC stage (Supplementary Table S2).”

Please present full name of BCLC when it first occurred.

7. Figure 7

Knockdown of lncGCLC significantly upregulated miR-122-5p expression. This is not shown in the Fig. 7.

8. Please read and carefully check through all the manuscript text, tables and figures. It is the authors’ responsibility to present their best work to the readers.

Author Response

Dear Reviewer:

Thank you for your comments concerning our manuscript entitled “Down-regulation of lncRNA GCLC-1 promotes microcystin-LR-induced malignant transformation of human liver cells through regulating GCLC expression” (Manuscript ID: toxics-2143435).

The comments are all valuable and helpful for us to revise and improve our manuscript. We have studied these comments carefully and have made some modification and corrections accordingly. Here we provide our responses, point by point, to the comments as listed below and highlighted the changes we have made in the revised manuscript.

We hope that the manuscript has been revised satisfactorily and will be acceptable for publication.

With best wishes,

Yours sincerely,

Qingqing Nong, MD, PhD

Professor of Preventive Medicine

School of Public Health

Guangxi Medical University

Nanning, 530021, P.R.China

Responds to the reviewer’s comments:

Response to Reviewer 1

Comments and Suggestions for Authors:

This manuscript aimed to study the role of lncGCLC pathway in MCLR-induced malignant transformation. Assays of knockdown of lncGCLC, cell proliferation, invasion and migration, xenograft nude mice model, human HCC tissues, as wells as GCLC expression and GSH levels, oxidative DNA damages were performed. The revised manuscript improved a lot. I still have the following comments and suggestions for the authors to improve the quality of the manuscript.

Response: Thanks for your encouraging and helpful comments. We tried our best to make some changes in order to improve the quality of the manuscript. The language errors throughout the whole paper have been corrected and marked in red in the revised manuscript.

  1. Section “2. Materials and methods”

Lines 121-122“Tissue samples were used to detect the expression of lncGCLC using qRT-PCR.” Please change “the expression of lncGCLC” to “the expressions of lncGCLC and GCLC”.

Response: Thank you for your suggestion. We have changed “the expression of lncGCLC” to “the expressions of lncGCLC and GCLC”, and marked it in red in the revised manuscript. (Materials and methods section, line 121, page 3)

  1. Lines 124-127

“The 30 participants were matched by age, sex, and hepatitis B virus-infection status, and were divided into two groups according to the median values of serum MCs (0.14 μg/L), termed the low MCs exposure group and the high MCs exposure group, respectively.”

The 30 participants were divided into two groups according to the median values of serum MCs (0.14 μg/L). However, in the supplementary table S2, numbers of participants in termed the low MCs exposure group and the high MCs exposure group were 13 and 17, respectively. If according to the median values, then numbers in the two groups should be both 15. Please check the data.

Response: Thank you for pointing this out. We are very sorry that we did not check it clearly. The 30 participants were divided into two groups according to the median values of serum MCs (0.14 μg/L), the numbers in the two groups should be both 15. Now, we have re-calculated and corrected the data in the revised Supplementary Table S2.

  1. Lines 183-185

“2.9. Tumorigenicity assays in nude mice

Thirty-six BALB/c athymic nude mice (4-5 weeks, male: female = 1:1) were purchased from the Guangxi Medical University Laboratory Animals Center.”

Figure 4, C-G

Both males and females were studied in this research. Were there any sex-dependent differences on tumor weight or volume? Please present both results of males and females.

Response: Thank you for your suggestions. We are very sorry that our analysis of the data is not comprehensive enough. According to your suggestion, we compared the tumor weights or volume between male and female nude mice in each group after inoculation for 22 days. No sex-dependent differences on tumor weight or volume were found. We also explained the experimental results and marked them in red in the revised manuscript. (Results section, line 329-331, page 9).

“Though the tumors tended to grow faster in male compared to female mice in the group injected with MCLR-transformed cells, the differences did not reach statistical significant level (Supplementary Figure S1).”

  1. Lines 185-187

“After a week of adaptive feeding in a pathogen-free room, mice were randomly divided into 6 groups (n=6 per group).”

It should be n=3 per group, for both males and females. Right? Please make the revisions in the revised manuscript.

Response: Thank you for your suggestions. We apologize that this sentence is not clear. Now, we have corrected it and marked it in red in the revised manuscript. (Materials and methods section, line 184-186, page 4 and Results section, line 339-340, page 9).

“After a week of adaptive feeding in a pathogen-free room, mice were randomly divided into 6 groups (n = 3 for both male and female nude mice per group).”

  1. Figures 1CD, 5CDE

Please present number of participants in each group in the revised manuscript.

Response: Thank you for your suggestions. Now, we have presented the number of participants in each group, and marked them in red in the revised Figure legend. (Results section, line 250-254, page 6 and line 390-395, page 11).

  1. Lines 242-244

“There was no significant difference between the groups divided by the clinicopathologic factors, including tumor size, differentiation, lymph node metastasis, and BCLC stage (Supplementary Table S2).”

Please present full name of BCLC when it first occurred.

Response: Thank you for your suggestions. We have added the full name of BCLC (Barcelona Clinic Liver Cancer) in the revised manuscript and marked it in red in the revised manuscript. (Results section, line 242-245, page 5)

“As shown in Supplementary Table S2, there was no significant difference between the groups divided by the clinicopathologic factors, including tumor size, differentiation, lymph node metastasis, and Barcelona Clinic Liver Cancer (BCLC) stages.”

  1. Figure 7

Knockdown of lncGCLC significantly upregulated miR-122-5p expression. This is not shown in the Fig. 7.

Response: Thank you for pointing this out. We are very sorry that we did not check it clearly.  As shown in the revised Figure 7, we drew a shape that looked like a capital "T" to show “Knockdown of lncGCLC significantly upregulated miR-122-5p expression”.

  1. Please read and carefully check through all the manuscript text, tables, and figures. It is the authors’ responsibility to present their best work to the readers.

Response: Thank you for your reminder. After several checks through all the manuscript text, tables, and figures, we found that there were still some language errors in it. We have corrected it in the revised manuscript and marked it in red. (Materials and methods section, line 186-188, page 4)

Thanks very much for your constructive comments and suggestions. On behalf of my co-authors, we would like to express our great appreciation to you.

Reviewer 2 Report

All comments have been addressed. Thank you.

Author Response

Dear Reviewer:

Thank you for your comments concerning our manuscript entitled “Down-regulation of lncRNA GCLC-1 promotes microcystin-LR-induced malignant transformation of human liver cells through regulating GCLC expression” (Manuscript ID: toxics-2143435).

The comments are all valuable and helpful for us to revise and improve our manuscript. We have studied these comments carefully and have made some modification and corrections accordingly. Here we provide our responses, point by point, to the comments as listed below and highlighted the changes we have made in the revised manuscript.

We hope that the manuscript has been revised satisfactorily and will be acceptable for publication.

With best wishes,

Yours sincerely,

Qingqing Nong, MD, PhD

Professor of Preventive Medicine

School of Public Health

Guangxi Medical University

Nanning, 530021, P.R.China

Responds to the reviewer’s comments:

Response to Reviewer 2

Comments and Suggestions for Authors:

All comments have been addressed. Thank you.

Response: Thanks for your encouraging and positive comments. We tried our best to make some changes in order to improve the quality of the manuscript. The language errors throughout the whole paper have been corrected and marked in red in the revised manuscript.        

Thanks very much for your constructive suggestions. On behalf of my co-authors, we would like to express our great appreciation to you.

Round 3

Reviewer 1 Report

Journal: Toxics (ISSN 2305-6304)

Manuscript ID: toxics-2143435-peer-review-v3

Type: Article

Title: Down-regulation of lncRNA GCLC-1 promotes microcystin-LR-induced malignant transformation of human liver cells through regulating GCLC expression

Topic: Environmental Toxicology and Human Health

This manuscript aimed to study the role of lncGCLC pathway in MCLR-induced malignant transformation. Assays of knockdown of lncGCLC, cell proliferation, invasion and migration, xenograft nude mice model, human HCC tissues, as wells as GCLC expression and GSH levels, oxidative DNA damages were performed. The revised manuscript improved a lot. However, there are still some issues to be addressed. I have the following comments and suggestions for the authors to improve the quality of the manuscript.

1. Supplementary table S2

Values in the present version were not consistent with those in the previous version. For example,

Male, 9+10=19, previously: 8+13=21;

Age (years) ˃ 50: 6+8=14, previously: 4+8=12;

HBV infection, 10+11=21, previously: 10+15=25;

Please carefully check the data. The values should be the same, regardless of the division of groups.

Please also check data in figures 1CD, 5CDE.

2. Supplementary figure S1

Please present results of statistical differences among treatment groups, similar to Fig. 4EF.

3. Please read and carefully check through all the manuscript text, tables, figures and supplementary files. It is the authors’ responsibility to present their best work to the readers.

Author Response

Dear Reviewer:

Thank you for your comments concerning our manuscript entitled “Down-regulation of lncRNA GCLC-1 promotes microcystin-LR-induced malignant transformation of human liver cells through regulating GCLC expression” (Manuscript ID: toxics-2143435).

The comments are all valuable and helpful for us to revise and improve our manuscript. We have studied these comments carefully and have made some modification and corrections accordingly. Here we provide our responses, point by point, to the comments as listed below and highlighted the changes we have made in the revised manuscript.

We hope that the manuscript has been revised satisfactorily and will be acceptable for publication.

With best wishes,

Yours sincerely,

Qingqing Nong, MD, PhD

Professor of Preventive Medicine

School of Public Health

Guangxi Medical University

Nanning, 530021, P.R.China

Responds to the reviewer’s comments:

Response to Reviewer 1

Comments and Suggestions for Authors:

This manuscript aimed to study the role of lncGCLC pathway in MCLR-induced malignant transformation. Assays of knockdown of lncGCLC, cell proliferation, invasion and migration, xenograft nude mice model, human HCC tissues, as wells as GCLC expression and GSH levels, oxidative DNA damages were performed. The revised manuscript improved a lot. However, there are still some issues to be addressed. I have the following comments and suggestions for the authors to improve the quality of the manuscript.

  1. Supplementary table S2

Values in the present version were not consistent with those in the previous version.

For example,

Male, 9+10=19, previously: 8+13=21;

Age (years) ˃ 50: 6+8=14, previously: 4+8=12;

HBV infection, 10+11=21, previously: 10+15=25;

Please carefully check the data. The values should be the same, regardless of the division of groups.

Please also check data in figures 1CD, 5CDE.

Response: Thank you for your comments.

We apologize that we did not write the sentences clearly in our revised manuscript dated Jan 19 and Jan 31. Now, we have corrected them and marked them in red in the revised manuscript. (Materials and methods section, line 117 and line 125, page 3).

The cutoff value of serum MCs (0.14 μg/L) was actually calculated according to the median value of serum MCs from all 541 HCC patients, rather than from the 30 HCC patients. Thus, among these 30 patients whose HCC tissues samples were available to collect and analyze, there were 17 patients with serum MCs higher than 0.14 μg/L (termed as high exposure group) while 13 patients with serum MCs lower than 0.14 μg/L (termed as low exposure group).

The data of Supplementary Table S2 in the version submitted on January 19 were correct.  We checked again, the data in Figures 1CD and 5CDE were correct.

When we got the comment suggesting us to divide the 30 patients equally into two groups (15 for each), we made a mistake unfortunately. We tried to follow the suggestion but misused the dataset. The data of Supplementary Table S2 in the version submitted on January 31 were not consistent with those in the version submitted on January 19. We apologize for this. As shown below, we resubmitted Supplementary Table S2.

  1. Supplementary figure S1

Please present results of statistical differences among treatment groups, similar to Fig. 4EF.

Response: Thank you for your suggestions.

Now, we added the results of statistical differences among treatment groups. As shown in Supplementary Figure S1, the tumors of the male or female mice of sh-lncGCLC+MCLR group were significantly larger than those of the sh-lncGCLC group (all P < 0.05). In both male and female mice, though the tumors tended to grow faster in the sh-lncGCLC+MCLR group compared to the sh-NC+MCLR group, but the differences did not reach statistical significant level (P > 0.05). As shown below, we resubmitted Supplementary Figure S1.

  1. Please read and carefully check through all the manuscript text, tables, figures and supplementary files. It is the authors’ responsibility to present their best work to the readers.

Response: Thank you for your helpful suggestions.

We have tried our best to check through all the manuscript and make some changes in order to improve the quality of the manuscript. The language errors throughout the whole paper have been corrected and marked in red in the revised manuscript.      (Result section, line 253-254, page 6 and line 394-395, page 11).

Round 4

Reviewer 1 Report

Journal: Toxics (ISSN 2305-6304)

Manuscript ID: toxics-2143435-peer-review-v4

Type: Article

Title: Down-regulation of lncRNA GCLC-1 promotes microcystin-LR-induced malignant transformation of human liver cells through regulating GCLC expression

Topic: Environmental Toxicology and Human Health

This manuscript aimed to study the role of lncGCLC pathway in MCLR-induced malignant transformation. Assays of knockdown of lncGCLC, cell proliferation, invasion and migration, xenograft nude mice model, human HCC tissues, as wells as GCLC expression and GSH levels, oxidative DNA damages were performed.

Thank the authors for the revisions! Congratulations to all the authors! Your manuscript can be accepted for publication in the journal Toxics!